# The Antiarrhythmic Activity of Novel Pyrrolidin-2-one Derivative S-75 in Adrenaline-Induced Arrhythmia

**DOI:** 10.3390/ph14111065

**Published:** 2021-10-20

**Authors:** Klaudia Lustyk, Kinga Sałaciak, Paula Zaręba, Agata Siwek, Jacek Sapa, Karolina Pytka

**Affiliations:** 1Department of Pharmacodynamics, Faculty of Pharmacy, Jagiellonian University Medical College, Medyczna 9, 30-688 Krakow, Poland; kinga.salaciak@doctoral.uj.edu.pl (K.S.); jacek.sapa@uj.edu.pl (J.S.); 2Department of Physiochemical Drug Analysis, Faculty of Pharmacy, Jagiellonian University Medical College, Medyczna 9, 30-688 Krakow, Poland; paula.zareba@uj.edu.pl; 3Department of Pharmacobiology, Faculty of Pharmacy, Jagiellonian University Medical College, Medyczna 9, 30-688 Krakow, Poland; agat.siwek@uj.edu.pl

**Keywords:** arrhythmia, antiarrhythmic effect, adrenaline-induced arrhythmia, α_1_-adrenoceptor, α_1_-adrenolytics, pyrrolidin-2-one

## Abstract

Arrhythmia is a quivering or irregular heartbeat that can often lead to blood clots, stroke, heart failure, and other heart-related complications. The limited efficacy and safety of antiarrhythmic drugs require the design of new compounds. Previous research indicated that pyrrolidin-2-one derivatives possess an affinity for α_1_-adrenergic receptors. The blockade of α_1_-adrenoceptor may play a role in restoring normal sinus rhythm; therefore, we aimed to verify the antiarrhythmic activity of novel pyrrolidin-2-one derivative S-75. In this study, we assessed the influence on sodium, calcium, potassium channels, and β_1_-adrenergic receptors to investigate the mechanism of action of S-75. Lack of affinity for β_1_-adrenoceptors and weak effects on ion channels decreased the role of these adrenoceptors and channels in the pharmacological activity of S-75. Next, we evaluated the influence of S-75 on normal ECG in rats and isolated rat hearts, and the tested derivative did not prolong the QT_c_ interval, which may confirm the lack of the proarrhythmic potential. We tested antiarrhythmic activity in adrenaline-, aconitine- and calcium chloride-induced arrhythmia models in rats. The studied compound showed prophylactic antiarrhythmic activity in the adrenaline-induced arrhythmia, but no significant activity in the model of aconitine- or calcium chloride-induced arrhythmia. In addition, S-75 was not active in the model of post-reperfusion arrhythmias of the isolated rat hearts. Conversely, the compound showed therapeutic antiarrhythmic properties in adrenaline-induced arrhythmia, reducing post-arrhythmogen heart rhythm disorders, and decreasing animal mortality. Thus, we suggest that the blockade of α_1_-adrenoceptor might be beneficial in restoring normal heart rhythm in adrenaline-induced arrhythmia.

## 1. Introduction

Cardiovascular diseases are the leading cause of death worldwide—around 17.9 million people die each year from coronary heart disease, cerebrovascular disease, rheumatic heart disease, and other conditions [1]. One of the most severe cardiac disorders is arrhythmia. We distinguish several classifications of arrhythmia, considering either its origin (ventricular or supraventricular) or its effect on the heart rate (tachycardia or bradycardia) [2]. Despite the high prevalence of this heart disease, we still do not have effective and safe antiarrhythmic drugs. All available drugs lead to more or less pronounced proarrhythmic activity manifested by the occurrence of *torsades de pointes*, the increased number of premature beats, subsequent beats, or reentry disorders [3,4,5,6]. Any compound that increases the duration of the action potential may lead to the development of ventricular tachycardia, and any compound that changes the conduction rate or the refractory period may increase the recurrence of arrhythmias [7]. Either the elderly, or patients with an impaired heart muscle contractility or people with electrolyte disturbances are susceptible to arrhythmogenic effects of antiarrhythmic drugs [8]. Thus, there is still an urgent need to search for novel compounds with better safety profiles that would effectively stop arrhythmia attacks.

All subtypes of α_1_ receptor are expressed in the cardiomyocytes and play an important role in heart functioning [9]. The activation of these receptors mediates physiologic or adaptive hypertrophy in both in vitro [10,11,12,13,14] and in vivo [15,16,17] studies, prevents cardiac myocyte death [18,19,20,21], augments the contractility [22,23,24], and induces ischemic preconditioning [25,26,27,28]. Moreover, the α_1_ receptors may be involved in the etiology of the ischemia- and reperfusion-related arrhythmias [29,30]. Furthermore, α_1_ receptors antagonists such as prazosin and phentolamine showed antiarrhythmic properties in ischemia-induced arrhythmias in animal models [31,32,33,34,35]. In addition, many novel compounds targeting α_1_ receptors possess a strong antiarrhythmic activity e.g., in the adrenaline- or barium chloride-induced model of arrhythmia or in the rat coronary artery ligation-reperfusion model, which is due to the α_1_-adrenolytic properties [36,37,38].

Numerous studies have shown that pyrrolidin-2-one derivatives have significant antiarrhythmic and hypotensive properties [39,40,41,42,43,44,45,46]. This effect is possibly mediated via α_1_ receptor blockade as these derivatives bind strongly to adrenergic receptors, especially when they have arylpiperazine fragment and hydrogen bond acceptor [47,48,49]. In the previous studies, we showed that S-75, a novel arylpiperazine derivative of pyrrolidine-2-one, possesses a high affinity toward α_1_ (p*K_i_* = 7.3) and α_2_ (p*K_i_* = 6.01) receptors [50]. Moreover, it turned out that the tested compound acts as an antagonist of both α_1A_ and α_1B_ receptor subtypes (EC_50_ equals 138.2 ± 21.9 nM and 2.53 ± 0.5 nM, respectively) [50]. Considering its receptor profile, S-75 has already been tested for its potential hypotensive properties—it significantly decreased the blood pressure at doses 5, 1, and 0.5 mg/kg in normotensive rats.

As the antiarrhythmic effect of S-75 has never been tested, here, we examined whether this compound prevented the occurrence of an abnormal heart rhythm in adrenaline-, calcium chloride-, and aconitine-induced arrhythmia and reversed the adrenaline-caused irregularities of the heart rhythm. Moreover, we evaluated the potential antiarrhythmic properties of S-75 in post-reperfusion arrhythmia in the isolated rat’s heart. To present a complete picture of how S-75 affects heart functioning, we investigated the influence of the novel tested compound on ECG both in vivo and ex vivo (in the isolated heart). In addition, to obtain a detailed pharmacological profile, we assessed the affinity of S-75 toward β_1_ receptors and sodium, potassium, and calcium channels.

## 2. Results

### 2.1. S-75 Showed Low Affinity toward Sodium, Potassium, and Calcium Channels

S-75 possesses low affinity toward sodium, potassium, and calcium channels. Moreover, the tested compound did not bind to β_1_ receptors (Table 1).

### 2.2. S-75 Prolonged the PQ and Decreased Heart Rate in the Normal ECG in Rats

S-75 given at dose 5 mg/kg had no influence on QRS (Column factor: F (3,15) = 1.611, ns; Row factor: F (5,15) = 3.184, *p* < 0.05), QT_c_ (Column factor: F (3,15) = 1.312, ns; Row factor: F (5,15) = 18.86, *p* < 0.0001) and QT (Column factor: F (3,15) = 1.095, ns; Row factor: F (5,15) = 3.297, *p* < 0.05) (Table 2). After 15 min the tested compound prolonged the PQ interval by 12.2% (Column factor: F (3,15) = 6.116, *p* < 0.01; Row factor: F (5,15) = 5.442 *p* < 0.01), but also decreased the heart rate by 7.4% (after 10 min) and 8.6% (after 15 min) (Column factor: F (3,15) = 9.272, *p* = 0.001; Row factor: F (5,15) = 82.7, *p* < 0.0001) (Table 2).

### 2.3. S-75 Prolonged the PQ and RR Interval in an Isolated Rat’s Heart

S-75 at dose 10^−8^ mol/L significantly prolonged the PQ interval by 15.1% (Column factor: F (3,9) = 22.55, *p* < 0.001; Row factor: F (3,9) = 0.5644, ns) (Figure 1A) and the RR interval by 10.5% (Column factor: F (3,9) = 12.19, *p* < 0.01; Row factor: F (3,9) = 0.8253, ns) (Figure 1D), not affecting either QRS complex (Column factor: F (3,9) = 2.738, ns; Row factor: F (3,9) = 2.633, ns) (Figure 1B) or QTc interval (Column factor: F (3,9) = 3.43, ns; Row factor: F (3,9) = 1.281, ns) (Figure 1C).

### 2.4. S-75 Showed Prophylactic Antiarrhythmic Properties in Arrythmia Models Caused by Adrenaline, but Not Calcium Chloride and Aconitine

The intravenous injections of either adrenaline (20 μg/kg), calcium chloride (140 mg/kg) or aconitine (20 μg/kg) caused atrioventricular disturbances and ventricular and supraventricular extrasystoles (Table 3).

The administration of the tested compound at dose 0.5 mg/kg before the arrhythmogenic substance (adrenaline) prevented or decreased the number of extrasystoles and atrioventricular blocks (ED_50_ = 0.05 (0.02–0.11). ED_50_ value for carvedilol was 0.36 (0.16–0.8).

S-75 did not significantly reduce the amount of calcium chloride- and aconitine-induced heart rhythm disturbances and showed no activity is these models of arrhythmia.

### 2.5. S-75 Did Not Possess the Antiarrhythmic Activity in the Post-Reperfusion Model in Isolated Rat Heart

S-75 did not show the activity in the post-reperfusion model of arrhythmia and the arrhythmia severity index was higher compared to the control group (F (4,15) = 6.854, *p* < 0.01) (Table 4).

### 2.6. S-75 Showed Therapeutic Antiarrhythmic Properties in Adrenaline-Induced Arrythmia Model

S-75 significantly decreased the number of adrenaline-induced extrasystoles by 72.6% compared to the vehicle-treated group (H (3,18) = 11.93, *p* < 0.001) (Figure 2). The reference compound, carvedilol, diminished the number of extrasystoles by 95.2%.

S-75 also diminished the number of bradycardia and block episodes in adrenaline-treated rats and this effect was more pronounced than that for carvedilol (Table 5).

## 3. Discussion

We showed for the first time that the studied pyrrolidine-2-one derivative S-75 possessed prophylactic and therapeutic antiarrhythmic activity in the model of adrenaline-induced arrhythmia. Furthermore, its prophylactic antiarrhythmic activity was stronger than that of carvedilol—a commonly used β-blocker with α_1_-adrenolytic properties [53]. However, S-75 did not prevent arrhythmia caused by either aconitine or calcium chloride administration, nor did it abolish the heart rhythm abnormalities in the post-reperfusion model of arrhythmia.

Pyrrolidyn-2-one derivatives have been shown to interact with adrenergic receptors [39,45,54]. Due to the highest affinity for α_1_-adrenergic receptors assessed in our previous study, we chose the compound S-75 for further experimentation [50]. Our functional experiments revealed that S-75 antagonized α_1A_- and α_1B_ receptors [50]. The α_1A_, α_1B_, and α_1D_ subtypes of α_1_-adrenoceptor have been found in the human heart. α_1D_ receptors play a marginal role, whereas α_1A_ mediate a positive inotropic response of atrial cardiomyocytes, and α_1B_ regulate the myocytes contractility [55]. Given previous in vitro studies demonstrated that S-75 blocked α_1A_- and α_1B_-adrenoceptors, and these receptors can contribute to the development of cardiac arrhythmias induced by catecholamines, we aimed to evaluate the compound’s antiarrhythmic activity.

As we mentioned already, commonly used antiarrhythmic drugs possess limited efficacy and safety. However, an ideal antiarrhythmic agent administered in an emergency should effectively stop an attack of arrhythmia, and if taken regularly, should prevent its recurrence. The most often used antiarrhythmic drugs nowadays are divided according to the Vaughan Williams classification. This classification is based on the mechanism of action of drugs and divides them into four classes: sodium channel blockers, β-blockers, potassium channel blockers, and calcium channel blockers [56]. Thus, as a first step we investigated whether other than α_1_ receptors might contribute to the potential antiarrhythmic effect of a novel compound—S-75.

In order to evaluate the effect of the tested compound on β_1_-adrenergic receptors, we performed the radioligand studies. Lack of affinity for β_1_-adrenergic receptors excluded the role of those adrenoceptors in the mechanism of action of S-75. Next, we decided to determine the effect of the tested derivative on ion channels. First, we investigated the influence of S-75 on voltage-gated sodium channels (site 2). The sodium channels in the cardiomyocytes’ sarcolemma enable the conduction of depolarizing sodium current in the 0 phase of the action potential [57]. The blockade of these channels is used in the pharmacotherapy of arrhythmias; however, despite the therapeutic value of the sodium channel blockers, their pro-arrhythmic activity should be kept in mind [58]. The affinity of S-75 toward sodium channel was about 3.5 times lower than veratridine used as a reference compound. Therefore, this ion channel can at least partially contribute to the pharmacological effect of the tested compound.

Next, we investigated whether S-75 targets the potassium hERG channel. The potassium channels encoded by the hERG gene play an important role in controlling the repolarization of ventricular cardiomyocytes. The blockage of potassium channels inhibits the rapidly activating delayed rectifier potassium current. As a result, the duration of an action potential is prolonged, which can induce the formation of early subsequent depolarizations and *torsades de pointes* [59]. Our experiments revealed that S-75 showed 1172 times lower affinity toward potassium channel hERG than withdrawn from market terfenadine, which can indicate a low potential of S-75 to cause *torsades de pointes* [60].

Finally, we evaluated the effect of S-75 on L-type calcium channels (verapamil site). Since Ringer’s first observation in 1883 that Ca^2+^ ions are essential for myocardial contraction, their function in the heart has become increasingly appreciated. Every action potential induced by Na^+^ influx through voltage-gated sodium channels causes Ca^2+^ flux through the voltage-activated L-type calcium channels [61]. However, the role of the calcium channels in the mechanism of action of S-75 is marginal since it showed 20 times lower affinity than the reference compound D-600.

Since many antiarrhythmic agents tend to possess life-threatening proarrhythmic potential, we assessed the influence of the studied compound on normal ECG in rats. Most importantly, S-75 did not prolong the QT_c_ interval, which may suggest no potential for the induction of ventricular tachyarrhythmias, in particular *torsades de pointes*. Instead, the tested compound prolonged the PQ interval by 12.2%, and this effect may be associated with slower conduction in the atrioventricular node [62]. S-75 also decreased the heart rate by 7.4% (after 10 min) and 8.6% (after 15 min), which can be linked to the blockade of α_1_ receptors. Our in vivo findings mostly align with the results obtained on the isolated rat heart. S-75 did not affect either QRS complex or QT_c_ interval, yet the dose of 10^−8^ mol/L significantly prolonged the PQ interval by 15.1%, which may be due to the interaction with L-type Ca^2+^channels. Moreover, the tested compound prolonged the RR interval by 10.5%, decreasing the heart rhythm and showing a negative chronotropic effect.

Although α_1_-adrenoceptor antagonists do not have a well-established position in the pharmacotherapy of arrhythmias, the α_1_-adrenergic blockade may be beneficial in restoring normal heart rhythm and may also reduce the risk of arrhythmia [63]. According to research conducted by Billman and colleagues in dogs after myocardial infarction, prazosin and a tested α_1_-receptor antagonist prevented an arrhythmia caused by cardiac ischemia, accordingly, in 78% and 93% of the animals [64]. In the study by Tölg et al [33], a non-selective α-adrenergic receptor antagonist (phentolamine) reduced the incidence of ventricular fibrillation in isolated ischemic rat hearts by 32%. In comparison, a selective α_1_ receptor antagonist (prazosin) decreased the risk of cardiac arrhythmias by 83% and a tested α_1A_-blocker (WB 4101) limited the incidence of arrhythmias by 49% [33].

Considering the α_1_-adrenolytic properties of S-75, we determined its antiarrhythmic activity in the adrenaline-induced model of arrhythmia. An arrhythmogen, adrenaline, induced arrhythmias manifested by extrasystoles, atrioventricular blocks, bradycardia, and mortality. S-75 showcased prophylactic antiarrhythmic activity in this model, reducing the amount of post-adrenaline extrasystoles, conduction blocks and decreasing mortality of rats. The tested compound possessed stronger prophylactic antiarrhythmic properties than carvedilol (a reference drug). Considering that carvedilol blocks both β_1_- and α_1_-adrenoceptors and presented weaker antiarrhythmic activity than S-75 in the adrenaline-induced model of arrhythmia, we think that the potent blockade of α_1_-adrenoceptors is essential to treat adrenaline-induced arrhythmia. The obtained results are promising, as a novel compound with such pharmacological properties can be used to prevent attacks of arrhythmia due to emotional or physical stress.

In the next stage of our research, we evaluated antiarrhythmic properties in arrhythmia models, i.e., aconitine- and calcium chloride-induced. Aconitine and calcium chloride induce the changes in intracellular ion concentration (accordingly Na^+^ and Ca^2+^), which subsequently leads to arrhythmia [65]. Our results indicate that S-75 did not significantly reduce the number of cardiac rhythm disturbances or animal mortality in both arrhythmia models. Consequently, the studies confirmed our in vitro results, and we may presume that the blockade of sodium or calcium channels was not responsible for the mechanism of the antiarrhythmic effect of S-75.

Subsequently, we assessed the activity of the tested compound in the post-reperfusion model in isolated rat heart. This model recreates the clinical situation of a patient with myocardial infarction. After restoring coronary flow in the previously ischemic area, as a result of metabolic changes (e.g., increased adrenergic stimulation, overproduction of reactive oxygen species, disturbance in calcium ion homeostasis) post-reperfusion arrhythmias develop [66]. In ischemic conditions, the number of α_1_-adrenergic receptors increases, and their activation induces heart rhythm disturbances [29]. Despite the effectiveness of α_1_-blockers in eliminating post-reperfusion arrhythmias, our tested compound was not effective in this model of arrhythmia. There are several explanations to why S-75 did not show an antiarrhythmic effect in this model but was effective in the adrenaline-induced arrhythmia. These arrhythmia models differ in etiology—in adrenaline-induced arrhythmia, the abnormalities in the heart rhythm result from adrenergic receptors’ stimulation, and in the post-reperfusion model, arrhythmia develops after ischemia, reactive oxygen species production, and inflammation. We speculate that S-75 may not be effective in the post-reperfusion model of arrhythmia due to the lack of anti-inflammatory properties–however, this hypothesis needs further confirmation. Moreover, we need to highlight that one model was performed in vivo and another, ex vivo, and this methodology difference might underlie the differences in results. Finally, the antiarrhythmic effect of S-75 in the post-reperfusion model might be observed in different dosage schemes.

Since S-75 showed prophylactic antiarrhythmic properties in the adrenaline-induced arrhythmia, we next investigated whether it demonstrates therapeutic antiarrhythmic activity (a situation where a tested compound is given immediately after the administering the arrhythmogen). An effective antiarrhythmic drug should restore normal sinus rhythm as quickly as possible when used in a patient during the attack of arrhythmia. S-75 showed significant therapeutic antiarrhythmic activity by reducing the amount of post-adrenaline extrasystoles, atrioventricular blocks, bradycardia, and decreasing animal mortality.

Limitations to our study include evaluating the antiarrhythmic activity of pyrrolidyn-2-one derivative S-75 only after a single administration. Assessing the ability of the tested compound to prevent the attacks of arrhythmia after the repeated administration would provide information about its mechanism of the antiarrhythmic effect and if S-75 possesses the potential of long-acting antiarrhythmic. Moreover, significant activity in the adrenaline-induced arrhythmia model may relate to antioxidant effects, which needs further investigation. Therefore, our future studies will focus on assessing the effects of prolonged administration of S-75 and evaluating its extensive pharmacological profile.

## 4. Materials and Methods

### 4.1. Drugs

The studied compound: 1-(4-(4-(2-chlorophenyl)piperazin-1-yl)butyl)pyrrolidin-2-one hydrochloride (S-75) was synthesized in the Department of Physiochemical Drug Analysis, Faculty of Pharmacy, Jagiellonian University Medical College [50]. The investigated compound was dissolved in saline and administered intravenously (*iv*).

Thiopental (Sandoz GmbH, Kundl, Austria) and propafenone (Polpharma, Starogard Gdański, Poland) were dissolved in saline and administered intraperitoneally (*ip*). Adrenaline (Polfa S.A., Warsaw, Poland), aconitine (Merck, Darmstadt, Germany), calcium chloride (P.O.Ch. S.A., Gliwice, Poland), amiodarone (Sanofi- Aventis, Paris, France), carvedilol (Sigma-Aldrich, Steinheim, Germany), and verepamil (Abbott, Abbot Park, IL, USA) were dissolved in saline and administered intravenously (*iv*). Heparin (Polfa S.A., Warsaw, Poland) was used as anticoagulant during experiments. The control groups received saline as a vehicle.

### 4.2. Animals

Male normotensive Wistar rats, weighing 200–250 g, were used in experiments. Animals were kept in groups of 3 in standard plastic cages (42.7 cm × 26.7 cm) at constant room temperature of 22 ± 2 °C, with 12:12 h light/dark cycle and ad libitum access to food and water. Six rats were used in each experimental and control groups. All injections were administered in a volume of 1 ml/kg. Immediately after each experiment, animals were killed by cervical dislocation. All conducted procedures were approved by the Local Ethics Committee for Experiments on Animals of the Jagiellonian University in Krakow, Poland.

### 4.3. Affinity for β_1_-Adrenergic Receptors

Radioligand binding assay was performed on tissue rat cortex. A volume of 50 µL working solution of the tested compound, 50 µL [^3^H]-CGP-12177 (with final concentration 0.9 nM) and 150 µL tissue suspension prepared in assay buffer (50 mM Tris-HCl, pH 7.6) were transferred to polypropylene 96-well microplate using 96-wells pipetting station Rainin Liquidator (Mettler Toledo, Columbus, USA). In order to measure the unspecific binding 1 μM of propranolol was applied. Microplate was covered with a sealing tape, mixed and incubated for 60 minutes at 37 °C. The incubation was terminated by rapid filtration through GF/B filter mate. Ten rapid washes with 200 µL 50 mM Tris buffer (4 °C, pH 7.6) were completed using 96-well FilterMate harvester (PerkinElmer, Boston, MA, USA). The filter mates were dried at 37 °C in forced air fan incubator and then solid scintillator MeltiLex was melted on filter mates at 90 °C for 4 min. The radioactivity on the filter was measured in MicroBeta TriLux 1450 scintillation counter (PerkinElmer, Boston, MA, USA). Data were fitted to a one-site curve-fitting equation with Prism 6 (GraphPad Software, San Diego, CA, USA) and K_i_ values were estimated from the Cheng−Prusoff equation [51].

### 4.4. Effect on Sodium, Potassium, and Calcium Channels

Radioligand binding assay for Na^+^ channel was carried out according to the method of Callaway et al., with slight modifications [67]. Weighed rat cerebral cortex was homogenized at concentration of 10% (*w*/*v*) in ice-cold 0.32 M sucrose, 10 mM phosphate buffer (pH 7.4) using an ULTRA TURRAX homogenizer and next centrifuged at 1000× *g* for 10 min (0–4 °C). Collected supernatant was centrifuged at 40,000× *g* for 45 min (0–4 °C) to obtain membrane fraction. The resulting pellet was resuspended in incubation buffer (containing (mM): 50 Tris base, 50 HEPES, 130 choline chloride, 5.4 KCl, 0.8 MgSO_4_, 5.5 glucose). A volume of 200 µL of the tissue suspension (300 µg/well) containing LQ scorpion venom (10 µg/well), 50 µL of [^3^H]-BTX solution (final concentration 10 nM, PerkinElmer, Boston, MA, USA) and 50 µL of the tested compound were incubated at 37 °C for 1 h. The reaction was terminated by rapid filtration over glass fiber filters (FilterMate B, PerkinElmer, Boston, MA, USA) using 96-well harvester (PerkinElmer, Boston, MA, USA). Eight rapid washes with 500 µL of ice-cold 50 mM Tris-HCl buffer, pH 7.4, were completed. Veratridine (300 μM) was used to define nonspecific binding. Filter mates were dried at 37 °C in forced-air incubator and solid scintillator (MeltiLex, PerkinElmer, Boston, MA, USA) was then melted on them at 100 °C for 5 min. The radioactivity on the filters was measured in MicroBeta TriLux 1450 scintillation counter (PerkinElmer, Boston, MA, USA). The compound was tested in a screening assay at final concentration of 100 µM and result was expressed as percent inhibition of [^3^H]-batrachotoxin binding.

The radioligand binding studies for K^+^ and Ca^2+^ channels were performed commercially by Eurofins Cerep (Celle l’Evescault, France) using testing procedures described elsewhere [68,69]. Briefly, radioligand binding assay for K^+^ channel was performed on human recombinant HEK-293 cells using [^3^H]-dofetilide as specific ligand and non-specific binding was determined in the presence of 25 μM terfenadine [69], whereas radioligand binding assay for Ca^2+^ channel (L, verapamil site) was carried out in the rat cerebral cortex using [^3^H]-D888 as a specific ligand and non-specific binding was determined in the presence of 10 μM D600 [68].

### 4.5. Effect on Normal Electrocardiogram in Rats

To eliminate the influence of the tested derivative on normal electrocardiogram (ECG), the experiment was performed. ECG tests were carried out with Aspel ASCARD apparatus (standard II lead, with the tape speed 50 mm/s and voltage calibration 1 Mv = 1 cm). The ECG was recorded prior and 5, 10, 15 min after *iv* administration of tested compound. The influence on PQ, QT, QT_c_ interval, QRS complex and heart rate (RR) was defined. The Bazzett’s formula: QT_c_ = QT/√RR was used to calculate QT_c_ [52].

### 4.6. Effect on Normal ECG of Isolated Rat Heart

Hearts form Wistar rats were isolated and immersed in Chenoweth-Koelle solution (120 mM NaCl; 5 mM KCl; 2.2 mM MgCl_2_; 19 mM KCl; 2.4 mM CaCl_2_; 10 mM glucose). Hearts were perfused according to the Langendorff model under a constant pressure of 70 cm H_2_O (6.87 kPa) with a solution at 37 °C, pH 7.4, constantly oxygenated (O_2_/CO_2_, 19:1) using the MyoHEART Langendorff System-900MH (Danish Myo Technology, Hinnerup, Denmark), which simultaneously monitored the ECG. After 20-min stabilization period, the tested compound was administered directly to the nutrient fluid every 15 min at increasing concentrations of 10^−10^–10^−8^. The influence on PQ, QT_c_ interval, QRS complex and heart rate (RR) was assessed. The Bazzett’s formula: QTc = QT/√RR was used to calculate QT_c_ [52].

### 4.7. Prophylactic Antiarrhythmic Activity in Adrenaline-, Aconitine-, and Calcium Chloride-Induced Arrhythmias

The experiments were carried out according to the method described by Szekeres and Papp (1975) [70]. The intravenous administration of adrenaline (20 μg/kg), aconitine (20 μg/kg) or calcium chloride (140 mg/kg) solution into the caudal vein of anesthetized rats (thiopental, 75 mg/kg) induced the heart rhythm disturbances. The studied compounds were injected *iv* 15 min before the arrhythmogen. The ECG was recorded during the first 2 min and in the 5th, 10th, and 15th min after the adrenaline, aconitine or calcium chloride injection. The criterion of antiarrhythmic activity was the decrease or complete absence of extrasystoles, atrioventricular blocks and bradycardia in the ECG recording in comparison with the control group (in case of aconitine- and calcium chloride-induced arrhythmia the absence of fibrillation was also taken into account). The ED_50_ (a dose producing a 50% inhibition of ventricular contractions) was calculated using the method of Litchfield and Wilcoxon [71]. The compound was administered at the dose 5 mg/kg, and we decreased the dose by half until the disappearance of antiarrhythmic activity.

### 4.8. Antiarrhythmic Activity in the Post-Reperfusion Model in Isolated Rat Heart

Hearts form Wistar rats were isolated and immersed in Chenoweth–Koelle solution (120 mM NaCl; 5 mM KCl; 2.2 mM MgCl_2_; 19 mM KCl; 2.4 mM CaCl_2_; 10 mM glucose). Hearts were perfused according to the Langendorff model under a constant pressure of 70 cm H_2_O (6.87 kPa) with a solution at 37 °C, pH 7.4, constantly oxygenated (O_2_/CO_2_, 19:1) using the MyoHEART Langendorff System-900MH (Danish Myo Technology, Hinnerup, Denmark), which monitored the ECG. After 20 min stabilization period, acute myocardial ischemia was induced locally by installing a clamp on the left coronary artery (ischemic period). After 15 min, the clamp was removed, and ECG changes were monitored for 30 min (reperfusion period). The tested compound was added to the perfusion solution 15 min before coronary artery clamping [39]. The point scale described by Bernauer [35] was used to assess the antiarrhythmic activity in this arrhythmia model. The occurrence of a specific type of heart rhythm disturbance during the reperfusion period was scored according to the following scale:occurrence of less than 10 extrasystoles—1 point;occurrence of more than 10 extrasystoles—2 points;presence of ventricular tachycardia—3 points;presence of fibrillation—4 points.

### 4.9. Therapeutic Antiarrhythmic Activity in Adrenaline-Induced Arrhythmia

The experiments were carried out according to the method described by Szekeres and Papp [70]. The heart rhythm disturbances were evoked by intravenous administration of adrenaline (20 µg/kg) to anesthetized rats (thiopental, 75 mg/kg). The tested compound was administered *iv* immediately after the injection of adrenaline, at dose 5 mg/kg. The ECG was recorded during the first 2 min and in the 5th, 10th, and 15th min after the adrenaline injection. The ECG was recorded for the first 2 min and then at 5, 10 and 15 min of the experiment. The criterion of antiarrhythmic activity was the decrease or complete absence of extrasystoles, atrioventricular blocks and bradycardia in the ECG recording in comparison with the control group [39].

### 4.10. Statistical Analysis

The number of animals in groups was based on our previous studies [50]. Results are presented as either means ± SEM or as a percentage of occurrence of specific events (like extrasystoles, fibrillations, tachy- or bradycardias, mortality). Where it was necessary, the normality of data sets and homogeneity of variance were determined using Shapiro–Wilk and Brown–Forsythe test, respectively. In our analysis, we used one-way repeated measures ANOVA followed by Dunnett’s post hoc (ECG measurements). The reported *p* values of all ANOVAs used the Geisser–Greenhouse correction when the sphericity assumption was not met. In cases when assumptions for normal distribution of data were not fulfilled (therapeutic arrhythmia), we used Kruskal–Wallis with Dunn’s post hoc test.

## 5. Conclusions

This work presents the pharmacological profile of pyrrolidine-2-one derivative S-75 and its effect on the cardiovascular system. The tested compound possessed prophylactic and therapeutic antiarrhythmic activity in adrenaline-induced arrhythmia, which is most likely mediated via α_1_-adrenoceptors. Our study indicates that S-75 can be a model structure for synthesizing compounds with the potential to treat or prevent arrhythmia attacks.

## Figures and Tables

**Figure 1 pharmaceuticals-14-01065-f001:**
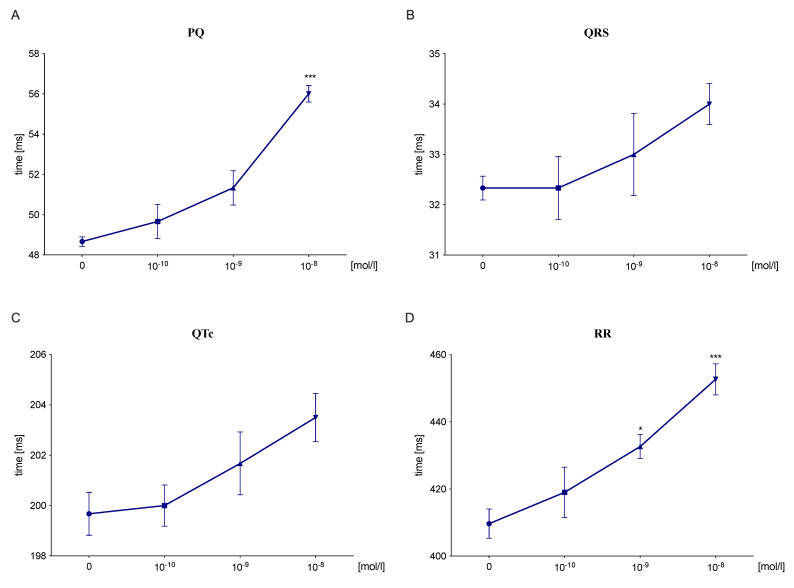
The effect of S-75 on the ECG components of the isolated rat heart. After stabilization period, the tested compound was administered directly to the nutrient fluid, in which heart was submerged, every 15 min at increasing concentrations of 10^−10^–10^−8^ mol/L. The influence on PQ (**A**), QRS complex (**B**), QTc (**C**), and RR interval (**D**) was assessed. The results are presented as mean ± SEM. Statistical analysis: Shapiro–Wilk test for data normality, and one-way repeated measures ANOVA (Dunnett’s post hoc); * *p* < 0.05, *** *p* < 0.001; *n* = 4 hearts.

**Figure 2 pharmaceuticals-14-01065-f002:**
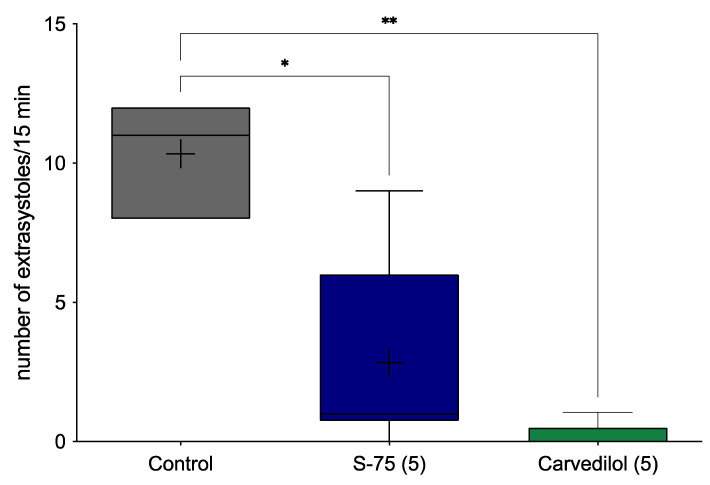
The therapeutic effect of S-75 in the adrenaline-induced arrhythmia model. After intravenous (*iv*) injection of adrenaline (20 µg/kg), the tested compound was immediately administered *iv* at dose 5 mg/kg. Control group received no additional treatment. The ECG was recorded for the first 2 min and then at 5, 10 and 15 min of the experiment. The criterion of antiarrhythmic activity was the decrease or complete absence of extrasystoles in the ECG recording in comparison with the control group. The results are presented as box plots showing the following data: mean (‘+’), median (horizontal line), upper and lower quartile (the width of the box shows interquartile range), upper and lower extreme (whiskers). Statistical analysis: Kruskal–Wallis test with Dunn’s post hoc; * *p* < 0.05, ** *p* < 0.01; *n* = 6 rats.

**Table 1 pharmaceuticals-14-01065-t001:** The affinity of S-75 for adrenergic β_1_ receptors, Na^+^, Ca^2+^ and K^+^ channels.

Treatment	β_1_ ^a,c^ (p*K_i_* [nM])	Na^+ a,d^ (IC_50_ [μM])	K^+^ hERG ^b,e^ (IC_50_ [μM])	Ca^2+ a,f^ (IC_50_ [μM])
S-75	n.c.	58	29.3	0.757
Propranolol	15.0	-	-	-
Veratridine	-	16.7	-	-
Terfenadine	-	-	0.025	-
D-600	-	-	-	0.037

Data are represented as p*K_i_*, that is, −log*K_i_* or IC_50_ and expressed as means from three independent experiments performed in duplicates. Inhibition constants (*K_i_*) and IC_50_ were calculated according to the equation of Cheng and Prusoff [51]. ^a^ Radioligand binding was performed using rat cortex tissue. ^b^ Radioligand binding was performed using HEK-293 cells. ^c^ The affinity values were determined using [^3^H]-CGP-12177. ^d^ The affinity values were determined using [^3^H]-batrachotoxin. ^e^ The affinity values were determined using [^3^H]-dofetilide. ^f^ The affinity values were determined using [^3^H]-D888.

**Table 2 pharmaceuticals-14-01065-t002:** Effects of an intravenous injection of the S-75 at dose 5 mg/kg on heart rate and ECG intervals.

Parameters	Time of Observation [min]			
	0	5	10	15
PQ	46.44 ± 1.352	47.00 ± 1.537	49.78 ± 1.042	52.11 ± 2.061 **
QRS	33.78 ± 1.384	35.56 ± 1.701	33.89 ± 1.628	37.22 ± 1.637
QTc	182.9 ± 4.805	192.3 ± 6.176	189.3 ± 9.543	190.9 ± 11.46
QT	81.57 ± 1.356	81.53 ± 1.812	82.47 ± 2.186	83.90 ± 1.730
Rate	310.8 ± 17.07	307.9 ± 18.93	287.8 ± 22.94 **	284.0 ± 23.50 **

Statistical analysis: Shapiro–Wilk test for normality, and repeated measures one-way ANOVA (Dunnett’s post hoc) ** *p* < 0.01; *n* = 6 rats. QT_c_– calculated QT interval according to Bazzett’s formula: QT_c_ = QT/√RR [52].

**Table 3 pharmaceuticals-14-01065-t003:** The effect of the tested compound on chosen parameters in arrythmia models in rats.

Treatment	Dose (mg/kg)	Fibrillations (%)	Extrasystoles (%)	Bradycardia (%)	Blocks (%)	Mortality (%)
Adrenaline-Induced Arrythmia						
Control	-	-	100	100	100	100
S-75	0.5	-	16.7	0	33.3	0
	0.25	-	66.7	16.7	50	0
	0.125	-	66.7	33.3	33.3	50
	0.06	-	83.3	100	50	0
	0.03	-	100	100	100	33.3
Carvedilol	0.25	-	100	16.6	100	66.7
Calcium Chloride-Induced Arrythmia						
Control	-	100	100	100	100	100
S-75	5	66.7	100	100	100	83.3
Verapamil	2.5	33.3	66.7	100	83.3	16.7
Aconitine-Induced Arrythmia						
Control	-	100	100	100	100	100
S-75	5	100	100	100	100	100
Propafenone	5	33.3	66.7	100	83.3	16.6

Except for propafenone (*ip*), the tested or reference compounds were administered intravenously (*iv*) 15 min before the experiment. Control group received no treatment except the administration of arrhythmogen. The observation was performed for 15 min after the *iv* injection of adrenaline (20 μg/kg), aconitine (20 μg/kg) or calcium chloride (140 mg/kg) i.e., during the first 2 min, at 5, 10, and 15 min. Results are presented as a percentage of the occurrence of specific events (fibrillations, extrasystoles, bradycardia, blocks, mortality). *n* = 5−6 animals.

**Table 4 pharmaceuticals-14-01065-t004:** The arrhythmic and antiarrhythmic activity of tested and reference compounds in post-reperfusion irregularities of heart rhythm.

Treatment	Concentration (M)	Extrasystoles (%)	Tachycardia (%)	Fibrillations (%)	Arrhythmia Severity Index
Control	-	100	100	33.3	6.3
S-75	10^−10^	100	100	100	9
Amiodarone	10^−9^	100	100	66.7	7.7
Propafenone	10^−11^	100	66.7	33.3	6.7
Quinidine	10^−6^	83.3	50	0	2.3 *

After stabilization period, acute myocardial ischemia was induced locally by installing a clamp on the left coronary artery (ischemic period) in the isolated rat heart. After 15 min, the clamp was removed, and ECG changes were monitored for 30 min (reperfusion period). The tested or reference compounds was added to the perfusion solution 15 min before coronary artery clamping. Control group received no treatment. Results are presented as a percentage of the occurrence of specific events (fibrillations, extrasystoles, bradycardia, blocks, mortality). Arrhythmia severity index was calculated according to Bernauer scale [35]. Statistical analysis: Shapiro–Wilk test for data normality, and one-way repeated measures ANOVA (Dunnett’s post hoc); * *p* < 0.05; *n* = 4 hearts.

**Table 5 pharmaceuticals-14-01065-t005:** The therapeutic antiarrhythmic effect of the tested compound and carvedilol in adrenaline-induced arrhythmia model.

Treatment	Dose (mg/kg)	Bradycardia (%)	Blocks (%)	Mortality (%)
Control	-	100	100	67.7
S-75	5	33.3	33.3	16.7
Carvedilol	5	67.7	83.3	16.7

The tested or reference compound was administered intravenously (*iv*) immediately after the injection of adrenaline, at dose 5 mg/kg. Control group received no treatment except the administration of arrhythmogen. The observation was performed for 15 min. Results are presented as a percentage of the occurrence of specific events (bradycardia, blocks, mortality). *n* = 5−6 animals.

## Data Availability

Data is contained within the article.

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
