# Peer review of "The Antiarrhythmic Activity of Novel Pyrrolidin-2-one Derivative S-75 in Adrenaline-Induced Arrhythmia"

_pharmaceuticals, 2021, doi:10.3390/ph14111065_

Round 1
Reviewer 1 Report
In this article, the authors investigated the antiarrhythmic activity of pyrrolidine-2-one derivative S-75 showed that it had prophylactic and therapeutic antiarrhythmic activity in the adrenaline-induced arrhythmia model because of its inhibitory action on α1-adrenoceptor but not β1-adrenoceptor. The experiments were carefully carried out, and the paper will provide useful information on α1-adrenoceptor antagonists as antiarrhythmic agents. My comments and questions are as follows:
Major comment
- Figure 1A: Why did S-75 prolong PQ interval in the isolated rat’s heart? In the Langendorff model, the activity of the autonomic nervous system might be minimal. S-75 had a higher affinity toward the L-type Ca2+ channel among the author’s tested channels. Consequently, it is possible that the prolonging effect of S-75 on the PQ and RR interval is mediated by the L-type Ca2+ channel but not by α1-adrenoceptor.
- Figure 1D: Which does figure 1D indicate heart rate or RR interval? If it indicates RR interval, the result of S-75 can be interpreted as a negative chronotropic effect.
Minor comments
- Page 4, line 127: Please describe the results in calcium- and aconitine-induced arrhythmia in the result section.
- Result section: Please correct the heading numberings.
Reviewer 2 Report
I consider that the idea of this study, to evaluate the antiarrhythmic activity of novel pyrrolidin-2-one derivative S-75, is very interesting. The article is well-written and comprehensive, with clear and legible tables and it may be accepted for publication. I consider that the findings are interesting and that the results obtained can make significant contributions to further large studies. However, I recommend emphasizing much better the original elements of their article.
